# Assessment of IL-6 Pathway Inhibition in Gastrointestinal Behçet’s Disease from Immunological and Clinical Perspectives

**DOI:** 10.3390/biomedicines13010247

**Published:** 2025-01-20

**Authors:** Makoto Naganuma, Mitsuhiro Takeno, Aykut Ferhat Çelik, Robert Moots, Philippe Pinton, Tadakazu Hisamatsu

**Affiliations:** 1Third Department of Internal Medicine, Kansai Medical University, Hirakata 573-1191, Japan; naganuma@hirakata.kmu.ac.jp; 2Department of Allergy and Rheumatology, Nippon Medical School Musashi Kosugi Hospital, Kawasaki 211-8533, Japan; m-takeno@nms.ac.jp; 3Cerrahpaşa Medical Faculty, Istanbul University-Cerrahpaşa, Fatih, Istanbul 34320, Türkiye; aykutferhatcelik@gmail.com; 4Faculty of Heath Social Care and Medicine, Edge Hill University, Ormskirk, Lancashire L39 4QP, UK; rjmoots@rheum4life.co.uk; 5Department of Rheumatology, Aintree University Hospital, Liverpool L9 7AL, UK; 6Clinical and Translational Sciences, Ferring Pharmaceuticals, 2770 Kastrup, Denmark; philippe.pinton@ferring.com; 7Department of Gastroenterology and Hepatology, Kyorin University School of Medicine, Tokyo 181-0004, Japan

**Keywords:** Behçet’s disease, biologics, inflammatory bowel disease, interleukin-6, olamkicept, signaling pathways, tocilizumab, trans-signaling

## Abstract

Behçet’s disease is an autoinflammatory disorder characterized by relapsing and remitting vasculitis that can manifest in various forms, including gastrointestinal Behçet’s disease (GIBD). Its complications (e.g., intestinal perforation) are among the primary causes of morbidity and mortality. GIBD pathogenesis involves the enhanced production of certain cytokines, e.g., tumor necrosis factor α and interleukin-6 (IL-6), which could serve as a target for potential therapies. This review provides an overview of GIBD, including the diagnosis and immunopathogenesis as it is currently understood, and evaluates the emerging role of the inhibition of IL-6 (classic and trans-signaling) as an alternative treatment option for patients with GIBD. Given the current paucity of data, we reflected on the potential of IL-6 inhibitors such as tocilizumab and olamkicept based on immunopathogenic considerations and available clinical data in patients with inflammatory bowel disease (IBD), in whom clinical response or remission was induced. The selective inhibition of IL-6 trans-signaling may bring new impetus to the development of this drug class, particularly regarding safety. Still, the benefits of IL-6 inhibitors for patients with GIBD need to be evaluated in appropriate proof-of-concept studies. The clinical outcomes of IL-6 inhibitors in IBD are promising and may suggest their potential relevance in GIBD.

## 1. Simple Summary

Behçet’s disease is a long-term (chronic) disease caused by an overreaction of the body’s immune system. This leads to the inflammation of blood vessels in different parts of the body. When blood vessels in the gut (stomach and intestine) are affected, gastrointestinal Behçet’s disease, or GIBD.

Individuals with GIBD may live with feeling sick (nausea), stomach pain, and diarrhea. GIBD can also cause a hole (perforation) in the gut, which can be life-threatening.

There are currently no medicines specifically designed to treat GIBD. In this review, we looked at published research on medicines called interleukin-6 (IL-6) inhibitors, already used to treat other conditions such as rheumatoid arthritis. These medicines shut off a substance released by cells of the immune system called IL-6. IL-6 can cause gut inflammation, but it can also help the body fight infections. We wanted to explore the potential of IL-6 inhibitors to help people with GIBD.

Studies have shown that IL-6 inhibitors can improve symptoms of other conditions affecting the gut, known as inflammatory bowel diseases. However, IL-6 inhibitors can increase a person’s risk of obtaining infections. Scientists have therefore developed more targeted (selective) IL-6 inhibitors that are less likely to increase the risk of infections.

Using what we know so far, we believe it is worth testing IL-6 inhibitors in clinical trials to see if they may help people with GIBD. Of particular interest is research using selective IL-6 inhibitors that aim to not increase the risk of infection.

## 2. Introduction

Behçet’s disease (BD) is an autoinflammatory disease characterized by relapsing and remitting vasculitis that can affect all parts of the body and lead to diverse clinical manifestations, typically including recurrent orogenital ulcerations, ocular (uveitis), and skin lesions [1,2]. Other manifestations include articular, vascular, neurological, and gastrointestinal involvement [1]. BD is particularly common in countries surrounding the Mediterranean Sea, the Middle East, and East Asia (China, Republic of Korea, and Japan), with the highest prevalence in Türkiye (20 to 420 cases per 100,000). A lower prevalence is observed in some northern European countries and the US [2,3]. In addition to genetic susceptibility (human leukocyte antigen B51 [HLA-B51], cytokines, etc.), environmental factors (e.g., infectious factors, dysbiosis) that trigger immune responses in BD are suggested to contribute to these geographic prevalence trends [4,5].

### 2.1. Gastrointestinal BD (GIBD)

Despite being less frequent than other symptoms in patients with BD, gastrointestinal manifestations and complications (e.g., massive bleeding or intestinal perforation) are among the primary causes of morbidity and mortality [6,7]. In patients with gastrointestinal involvement, lesions occur in the gastrointestinal tract, predominantly in the terminal ileum and ileocecal valve, causing symptoms that include abdominal pain, diarrhea, vomiting, weight loss, and bloody stool [2,6,7]. Data from a few studies indicate that between 13% and 27% of patients with GIBD experience intestinal perforations [7,8], which occur predominantly in the terminal ileum or the ileocecal region and ascending colon [8].

Patients with GIBD are more frequently diagnosed in distinct geographic regions, and the reported global prevalence varies significantly [6]. However, available data must be interpreted with caution, taking into account the different diagnostic criteria and methodology, distinct clinical disciplines involved (e.g., gastroenterology, dermatology), treatment history (e.g., non-steroidal anti-inflammatory drugs [NSAIDs]), and potential misdiagnoses [6]. Epidemiological studies, most with one or more of the above-mentioned limitations, suggest that between 15% and 50% of patients with BD in East Asia, notably Japan and Korea, may have gastrointestinal involvement [6]. The more realistic incidence may fall in the range of 15–25% in Japan, 10–15% in Korea, and less in the Middle East and Europe, approaching 1% in Türkiye. Proportions up to 40% of GIBD have also been reported in low-prevalence countries for BD, such as the US and the UK [6]. This high GIBD prevalence potentially indicates more frequent referrals due to gastrointestinal involvement compared with the less prominent mucocutaneous findings in these countries [6].

### 2.2. Diagnosis of GIBD

In general, diagnostic criteria for BD lack specificity for the diagnosis of gastrointestinal involvement. According to Japanese consensus statements first proposed in 2007, in addition to clinical findings that meet the diagnostic criteria for BD, one of the following criteria must apply: (A) a typical oval-shaped large ulcer in the terminal ileum or (B) ulcerations or inflammation in the small or large intestine [9]. In 2009, a group of Korean experts established an algorithm that includes both systemic clinical and endoscopic findings (Table 1) [10]. Because definitive GIBD criteria are time-dependent, patients who do not currently fulfill GIBD criteria must undergo further monitoring [10,11]. There are three consensus-based diagnostic categories: definite, probable, and suspected GIBD [10]. In 2020, evidence-based guidelines for GIBD developed by the Japanese Society for Behçet’s Disease suggested an algorithm for the diagnosis of GIBD that includes right lower abdominal pain, bloody stool, volcano-shaped ulcers around the ileocecal region, and exclusion of other differential diagnoses [12].

An endoscopic evaluation is usually performed to assess characteristic intestinal ulcerations in GIBD. Complementary imaging diagnostic tools include abdominal ultrasound imaging, computed tomography (CT) scan, and magnetic resonance enterography (MRE), which may provide useful information on intestinal wall thickening (e.g., to assess transmural healing) and inflammatory masses. These methods also help to exclude other pathologic abdominal conditions [6,12].

Because BD has characteristics of both vasculitis and inflammatory disease, diagnosis of GIBD can be challenging, as patients present with gastrointestinal and extra-intestinal clinical features similar to inflammatory bowel disease (IBD), particularly Crohn’s disease (CD) [2]. A diagnostic scoring system has been developed to differentiate between GIBD and CD [13]. The score is based on five parameters with respect to the ulcers: (1) shapes, (2) distributions, (3) numbers, and the presence of (4) cobblestone, and (5) aphthoid lesions [13]. For example, compared with volcano-shaped and deep ulcers observed in GIBD, ulcers in CD are longitudinal and occur on the mesenteric attachment side of the small intestine, with possible cobblestone appearance [13].

Fever occurs in 25% of patients with GIBD [14]. Up to 80% of patients with episodic fever, trisomy 8-positive myelodysplastic syndrome (MDS), and MDS-related transformations fulfill the criteria for BD diagnosis and have very similar gastrointestinal endoscopic ulcer morphology [15]. Fever also makes the exclusion of acute or chronic infections (e.g., intestinal tuberculosis) necessary [6].

Monogenetic IBD (e.g., chronic granulomatous disease, interleukin-10/interleukin-10 receptor deficiencies) is common in early childhood, often with fever, aggressive mucocutaneous findings, gastrointestinal involvement with perianal lesions and immunodeficiencies [16]. Because BD and autoinflammatory monogenetic conditions share similar features, there is an ongoing debate among BD specialists as to whether BD is an autoinflammatory disease. Lastly, rare upper-gastrointestinal involvement (e.g., esophageal, gastric, and duodenal) should be confirmed after exclusion of NSAID-induced ulcers, viral infections (e.g., cytomegalovirus, herpetic stomatitis, and esophagitis), and peptic ulcers (frequently proton-pump inhibitor-responsive) [6,17].

The severity of GIBD can be defined by a comprehensive assessment of symptoms, inflammatory response, intestinal ulcer findings, and degree of anemia [12]. The disease activity index of BD (DAIBD; scores between 0 and 325) may help to define the disease severity of GIBD, with scores of ≤19, 20–39, 40–74, and ≥75 corresponding to quiescent, mild, moderate, and severe disease, respectively [7]. However, further studies involving different ethnic groups are needed to validate the use of DAIBD to accurately predict disease course based on endoscopic findings and clinical symptoms [18].

### 2.3. Immunopathogenesis of BD

BD has been linked to increased activity of the innate and adaptive immune systems, triggered by genetic susceptibility (e.g., HLA-B51, cytokines, and others) and environmental factors (e.g., infectious agents) [19,20]. Carriers of the *HLA-B*51* allele, a gene variant with high prevalence in populations living in areas along the historic Silk Road, are particularly susceptible to BD [4,21]. Multiple studies, including genome-wide association studies, suggest that an association with BD is unlikely due to *HLA-B*51* alone but rather to a linkage with other variants in the major histocompatibility complex (MHC) locus [4]. The similarities between BD and MHC-I-associated spondyloarthropathies, anterior uveitis, and birdshot uveitis have been described [22] and include antigen processing genes endoplasmic reticulum aminopeptidases (*ERAP1* and *ERAP2*) and the interleukin-17 (IL-17) pathway gene interleukin-23 receptor (*IL-23R*), thus implicating MHC-I peptide presentation as a mechanism involved in these MHC-I-related spondyloarthropathies.

Characteristic immune responses in BD include neutrophil hyperactivity, imbalances between regulatory T-cells (Treg) and pro-inflammatory T-helper (Th) 1 and Th17 cells, as well as enhanced production of involved cytokines, e.g., IL-17, tumor necrosis factor (TNF)-α, and IL-6. Mechanisms and pivotal cytokines that stimulate T-cell differentiation towards pro-inflammatory subsets are shown in Figure 1 [20,23]. In addition to Th1/Th17 activation, the balance of Th17 and Treg cells is crucial in triggering inflammatory responses in patients with active BD. A proper balance between Th1/Th17 and Treg cells normally ensures effective immunity while preventing pathological autoimmunity. Furthermore, IL-6 and TNF-α enable differentiation into Th22 cells, which produce IL-22—this is another type of pro-inflammatory cytokine that particularly contributes to mucocutaneous lesions in patients with BD [20].

Environmental factors do not appear to directly cause BD but are thought to trigger an autoinflammatory response in individuals with a genetic susceptibility to BD [1,4]. The disease-associated nonsynonymous variants in the Mediterranean fever gene (*MEFV*) and toll-like receptor 4 (*TLR4*) implicate innate immune and bacterial sensing mechanisms in BD pathogenesis [24]. BD shows characteristics of autoimmune diseases with aberrant T- and B-cell responses, e.g., to heat-shock proteins (HSPs), endothelial cells, enolase, and retinal S antigen [5]. Bacterial and viral infectious by-products, such as *Streptococcus sanguinis* antigens or herpes simplex virus DNA, reported to have high homologies with human HSPs, are implicated in BD pathogenesis [21]. However, the overall mechanism of BD (immune) pathogenesis remains unclear, especially due to the lack of relevant animal models that reflect BD in its complexity and diversity, compounded by different mechanisms involved in individual manifestations [25].

### 2.4. Immunopathogenesis of GIBD

Considering that the pathogenesis of BD needs further research to be fully understood, it is not surprising that even less is known about the specific pathology of GIBD. While HLA-B51 is the predominant genetic susceptibility factor associated with BD, recent Asian epidemiological studies indicate that it is much less frequently associated with GIBD compared with ocular or cutaneous phenotypes [4,26]. However, whether these findings are limited to certain geographic regions remains unclear.

A few studies have tried to identify other genetic variations that might be linked to GIBD, such as single nucleotide polymorphisms (SNPs) of *IL-23R*, *IL-17A*, and signal transducer and activator of transcription 4 (*STAT4*), as well as *HLA-B*46:01* and two SNPs of tyrosine-protein phosphatase non-receptor type 2 (*PTPN2*) [27,28,29]. In patients with BD and early gastrointestinal involvement, both Th1 and Th17 cells infiltrating the intestinal mucosa were suggested to be key drivers of inflammation, causing mucosal damage via increased production of pro-inflammatory cytokines (e.g., IL-17, interferon [IFN]-γ), along with intestinal T-cells that produce large quantities of TNF-α [30]. These findings could provide a scientific rationale not only for targeting Th17-related cytokines (e.g., with anti-IL-17 agent secukinumab) but also for other biological agents that can inhibit Th17 differentiation and ‘restore’ T-cell balance to prevent severe gastrointestinal complications when used in early-stage disease [30]. These would include anti-TNF-α and potentially anti-IL-6 agents, as IL-6 promotes Th17 differentiation [20,30]. However, because only a few patients were included in this study [30], the role of Th1 and Th17 needs to be further elucidated in larger studies. Additional insights are expected from an ongoing clinical trial that aims to further elucidate the pathogenesis of GIBD [31].

In two cross-sectional, single-center studies conducted in China, enhanced levels of IL-6 (>7 pg/mL), among other inflammation markers, correlated with disease activity and were more associated with GIBD than mucocutaneous or ocular phenotypes [32,33]. Although still unclear, the current understanding of the immunopathogenesis and the inflammatory cytokines involved in GIBD provide a basis for the development of treatment modalities [20].

## 3. Overview of Approved and Emerging Treatments for GIBD

To date, BD cannot be cured; thus, treatment and management approaches focus on the prompt suppression of inflammatory exacerbations and recurrences to avoid irreparable organ damage [34]. In patients with GIBD, the induction and maintenance of clinical and endoscopic remission (endoscopic healing) are the ultimate goals of medical treatment [6]. Although endoscopic remission is more favorable than clinical remission in terms of prognosis, the risks associated with enhanced treatment to achieve this goal should be considered [12].

Given the heterogeneous nature of BD, current treatment strategies depend on observed clinical manifestations and their severity. Agents commonly used to treat IBD form the basis for current treatment strategies for GIBD. Induction therapies include 5-aminosalicylic acid (5-ASA) and derivatives such as sulfasalazine for mild-to-moderate and corticosteroids and anti-TNF-α agents for moderate-to-severe GIBD [35]. Moderate-to-severe disease activity with manifestations of severe systemic symptoms and recurrent gastrointestinal bleeding may lead to the use of corticosteroids as induction therapy. These are particularly effective in reducing ulcer size [6,12,36]. Of note, tapering of corticosteroid dose would be required for patients in remission, which often causes symptoms to recur [6,12]. Whether the effectiveness of high-dose corticosteroids in GIBD outweighs the risks, such as intestinal perforation, is still unclear [35,36]. Furthermore, the efficacy of corticosteroids, 5-ASA, and sulfasalazine for the treatment of GIBD has not been confirmed in adequately powered clinical trials [7,12]. Although colchicine is commonly used as a first-line treatment for other types of BD, such as mucocutaneous and articular BD, its efficacy as an induction or maintenance treatment in GIBD has not been formally established. Therefore, the use of colchicine as monotherapy for mucosal inflammation and ulcers related to GIBD is not recommended. It may be used occasionally in selected patients. Immunomodulators like azathioprine or methotrexate remain options in more severe or unresponsive cases [12,35,36].

In patients with moderate-to-severe GIBD who fail to respond to conventional systemic treatments, anti-TNF-α agents can be used [12,35]. For example, infliximab and adalimumab (both of which are approved for GIBD and covered by insurance in Japan) have demonstrated clinical efficacy in patients with severe and resistant GIBD (Table 2) [12,36,37,38,39,40]. The clinical evidence for the use of infliximab and adalimumab is largely based on retrospective and prospective single-arm studies with relatively small numbers of patients. Therefore, two systematic reviews with meta-analyses aimed to shed light on the overall therapeutic value of anti-TNF-α biological agents in the treatment of GIBD. Both studies confirmed the efficacy and acceptable safety profiles of these agents [41,42]. More recently, the real-world efficacy and safety of infliximab in patients with GIBD and other subtypes have been confirmed [43]. Adverse events (AEs) include autoimmune and cardiac conditions, malignancies, and infections (mild or serious), the latter being a significant concern, especially in the context of long-term use [44]. Anti-TNF-α agents can also be considered to reduce or avoid the use of corticosteroids [12]. Because of the unpredictable symptom flare-ups and the risk of GIBD symptom recurrence, patients may receive therapeutic anti-TNF-α following medical or surgical induction therapy [12].

Severe manifestations of GIBD are often associated with multiple relapses and a poor prognosis, particularly in patients requiring surgical intervention [6,7]. Surgery is indicated for patients who are refractory to pharmacological treatment and those with severe gastrointestinal complications, such as perforation and excessive bleeding; often, repeated surgical intervention is required [2]. Furthermore, initial response rates to medical treatment (clinical remission after eight weeks) range from 38% to 46% in patients who have undergone gastrointestinal surgery and reach 67% for nonsurgical patients [6]. In the postoperative setting, immediate medical treatment should be considered to minimize the risk of recurrence and reoperation. Postponement of surgery until the disease activity is under control is recommended [6,12]. Predictive factors for reoperation include volcano-shaped deep ulcers, corticosteroid use, and postoperative complications [12].

Beyond anti-TNF-α agents, there is little clinical evidence to date for the use of anti-IL-1 agents anakinra and canakinumab in GIBD. They may have potential in the management of GIBD, but these assumptions are based on observations in small subpopulations of patients with gastrointestinal symptoms from two case series reports and a retrospective cohort study [45]. Controversial observations have been made regarding the efficacy of the anti-IL-6 monoclonal antibody (mAb) tocilizumab in GIBD (discussed in greater detail in the following sections), and further investigation is required due to a risk of intestinal perforation [45,46]. Other emerging biological agents that have been shown to be effective in the management of certain BD manifestations, such as secukinumab (anti-IL-17) and ustekinumab (anti-IL-12/IL-23), have yet to be investigated in GIBD [23]. Other suggested treatment options for GIBD include the small molecule drugs baricitinib, a Janus kinase (JAK)1/JAK2 inhibitor, and the phosphodiesterase-4 inhibitor apremilast [47,48,49]. Baricitinib has shown favorable and glucocorticoid-sparing effects in a few patients with refractory GIBD [47]. Apremilast is approved for the treatment of oral ulcers associated with BD but has also been shown to improve gastrointestinal manifestations in some cases, either alone or in combination with anti-TNF-α agents [48,49]. Diarrhea is one of the most common AEs of apremilast, [50] which must be taken into account when considering its use in patients with GIBD. Overall, larger controlled studies are needed to identify and support the optimal management of patients with GIBD [35].

## 4. Rationale for the Potential Use of Anti-IL-6-Trans-Signaling Agents in GIBD

Since the advent of biological agents for the treatment of BD, much attention has been paid to anti-TNF-α agents; however, there is an increasing number of biologics that target more or other cytokines, including IL-1 and IL-6 [45].

The rationale for targeting IL-6 signaling to treat inflammatory disease has emerged as a logical consequence of its central role in the immune response [51,52]. Multiple anti-IL-6 agents, targeting either the IL-6 or the IL-6 receptor (IL-6R), are currently in clinical use or late-stage development. The first IL-6R-neutralizing mAb was approved in Japan in 2005 to treat Castleman’s disease, a lymphoproliferative disorder. Since then, siltuximab and tocilizumab have been approved for the same indication [51]. Tocilizumab also received approval for the treatment of several inflammatory diseases, such as rheumatoid arthritis (RA), juvenile inflammatory arthritis, and, more recently, giant cell arteritis and Takayasu arteritis [51,52,53,54].

IL-6 is a pleiotropic cytokine that is produced by several immune and non-immune cells and takes part in the regulation of the immune response, hemopoiesis, and organ function (e.g., nervous system, cardiovascular system, or liver). It is produced by various stimuli, such as bacterial and viral infections or cytokines, including TNF-α. IL-6 has long been recognized for its involvement in acute phase response to infection, inflammation, or tissue damage through stimulation of hepatocytes to secrete acute-phase proteins such as C-reactive protein (CRP), a diagnostic marker of inflammation and microbial infection. More recently, its central role in T-cell immunity—specifically, in balancing the differentiation of CD4+ T-cells—has been discovered [52,55,56,57]. Although IL-6 is often considered a pro-inflammatory cytokine, mechanistic studies have revealed a dichotomy between pro- and anti-inflammatory effects, which are facilitated by two distinct IL-6 signaling pathways (Figure 2) [51]: (1) the classic ligand-receptor pathway via membrane-bound IL-6R is responsible for anti-inflammatory and protective activities (e.g., infection defense) and (2) the trans-signaling pathway via circulating soluble IL-6R (sIL-6R), which triggers pro-inflammatory processes. Classically, IL-6 bound to IL-6R associates with the receptor subunit glycoprotein 130 (gp130) to initiate intracellular signaling via the Janus kinase (JAK)/STAT and rat sarcoma proto-oncogene (RAS)/mitogen-activated protein kinase (MAPK) and phosphoinositide-3 kinase (PI3K) pathways [58]. However, an sIL-6R can be generated through IL-6R cleavage by a disintegrin and metalloprotease 17 (ADAM17), whose activity is particularly enhanced during inflammation [51]. In the trans-signaling pathway, IL-6 can form an agonistic complex with sIL-6R, which binds to trans-membrane gp130 dimers present on a multitude of cell types that do not express membrane-bound IL-6R, leading to STAT induction in cells that do not normally respond to IL-6 [51]. Notably, trans-signaling potentially evokes higher amplitude and longer signaling compared with classic signaling by activation of all cellular gp130 proteins and slower and less efficient internalization of the IL-6/sIL-6R complex compared with activated membrane-bound IL-6R. The activity of the IL-6/sIL-6R complex is inhibited by high plasma levels of circulating soluble gp130 (sgp130), which effectively competes with membrane-bound gp130 [51].

There is evidence of a link between IL-6 activity and IBD pathogenesis, where inappropriate activation of the mucosal-associated immune system causes gastrointestinal inflammation and tissue damage [59]. The main cells involved are intestinal T-cells and macrophages, both of which produce increased amounts of IL-6 in patients with ulcerative colitis (UC) and CD [60]. In patients with CD, mucosal T-cells showed clear evidence for IL-6 trans-signaling and blockade-caused intestinal T-cell apoptosis [61]. Similarly to findings in IBD, enhanced expression and levels of IL-6 have been found in patients with active BD [62,63,64]. Recent studies in patients with BD concluded that IL-6 levels of >7 pg/mL (among other inflammation markers) correlate with disease activity and are particularly indicative of a GIBD phenotype [32,33]. IL-6 is released early in inflammatory processes and promotes T-cell growth and cytotoxic T-cell differentiation, e.g., into Th17 cells [62]. The overall pattern of increased T-cell activity, including differentiation to cytotoxic Th1 and Th17 cells promoted by IL-6, is a key immunopathogenic similarity between IBD and GIBD [65].

### 4.1. Efficacy and Safety of Anti-IL-6 Agents in IBD

Agents targeting IL-6 signaling have been investigated in patients with IBD due to the crucial role that IL-6 plays in intestinal inflammatory processes [60]. Several molecules inhibit both classic and trans-signaling of IL-6, but so far, only one biologic in clinical development exclusively blocks trans-signaling [58]. To date, no IL-6 inhibitors have been approved for the treatment of IBD.

The anti-IL-6R mAb tocilizumab effectively induced a clinical response in patients with active CD. However, results were less promising compared with anti-TNF-α mAbs, as no endoscopic or mucosal healing was observed (Table 3) [66,67]. Similarly, PF-04236921, an anti-IL-6 mAb, induced clinical response and remission in anti-TNF-α-refractory patients with moderate-to-severe CD (ANDANTE I study). Endoscopic or mucosal healing was not investigated in this study (Table 3) [68]. The ANDANTE II study raised concerns about the increased risk of gastrointestinal perforation and abscesses with anti-IL-6 agents in these patients [68]. Although no such cases occurred with tocilizumab in the CD studies [66], a higher risk of gastrointestinal perforation with tocilizumab was observed in several studies in patients with RA, primarily in patients with diverticulitis [69,70].

Serious AEs in patients with CD that led to discontinuation of tocilizumab included one case of intraperitoneal abscess in the placebo group and one case of paralytic ileus in the treatment group; the latter was possibly drug-related, according to investigator assessments [66]. PF-04236921 treatment led to serious infections in six cases [68]. Although no serious infections occurred in CD patients treated with tocilizumab, they were among the most frequently observed serious AEs in RA studies with tocilizumab [70]. Inhibition of the classic IL-6 signaling pathway is thought to be a factor in both increased risk of infection and gastrointestinal AEs due to hampered intestinal wound healing [52,69]. According to a systematic review and meta-analysis, the increased risk of serious (including opportunistic) infections with tocilizumab treatment in patients with RA seems to be in a similar range as for anti-TNF-α agents [73]. Whether IL-6 inhibition plays a general role in predisposition to (serious) opportunistic infections needs to be further elucidated [52].

Olamkicept specifically inhibits IL-6 trans-signaling. It was well tolerated and effective in inducing a clinical response and remission in patients with IBD (Table 3) [72]. Clinical response and remission onsets and rates were in a similar range to those observed for anti-TNF-α or other biologic agents [72]. Clinical effectiveness coincided with target inhibition (reduction in epithelial STAT3 phosphorylation, seen 4 h after infusion and throughout the entire treatment) and transcriptional changes indicative of mucosal healing (Table 3) [71]. The mucosal transcriptional remission signature of olamkicept seemed to differ from those observed with other IBD treatments, e.g., infliximab. In addition, the target engagement signature of olamkicept-mediated IL-6 trans-signaling inhibition differed from that of broader IL-6R inhibition by tocilizumab [71]. Intestinal perforation or serious infections, which were a concern in patients treated with other anti-IL-6 agents for a similar period [68,70], did not occur with olamkicept, although the study was not designed to assess safety [72]. Data from two phase 1 trials demonstrated safety and tolerability after single and multiple dosing of olamkicept in healthy subjects and patients with CD [74]. Taken together, these findings further suggest that olamkicept may offer a mode of action and biological responses that differentiate it from current treatment strategies [71,72].

### 4.2. Efficacy and Safety of Anti-IL-6 Agents in BD

Clinical results suggesting the effectiveness of tocilizumab in patients with GIBD are derived from very few patient cases [75,76]. For example, tocilizumab treatment resolved symptoms in one patient who was refractory to conventional treatment and intolerant of TNF-α inhibitors [75]. However, a systematic literature review found that tocilizumab was not effective in patients with BD and gastrointestinal involvement [76].

Two recent studies indicate the general effectiveness of tocilizumab in refractory patients with other BD phenotypes. A proportion of 67% and 60% of patients with uveitis and neurological manifestations, respectively, had a complete response (N = 30). Three patients presented with serious AEs, including one case of intestinal perforation [77]. Recent evidence suggests that tocilizumab may be similarly effective as anti-TNF-α agents in patients with ocular BD [78]. In a single-center, observational study in China, complete remission with tocilizumab was observed in six of ten patients with refractory arterial lesions. No treatment-related AEs were reported [79]. In both studies, the use of tocilizumab led to a significant reduction in corticosteroid doses [77,79].

The approval of tocilizumab in giant cell arteritis, a large vessel vasculitis [53], may further promote its investigation in BD, as both diseases belong to the group of vasculitides. Further clinical evaluation of tocilizumab in patients with refractory BD, including GIBD, is needed. However, two phase 2 trials have been prematurely terminated, one due to low enrolment (NCT01693653) and the other due to primary and secondary outcome measures in three consecutive patients with BD uveitis (NCT03554161) [80,81].

### 4.3. Opportunities and Potential of Anti-IL-6 Agents for the Treatment of GIBD

There is a high unmet need to manage BD, particularly those phenotypes with higher mortality rates such as GIBD. Patients may not respond to current therapies, or only temporarily before they become refractory, and may therefore receive multiple types of drugs during their disease [18]. Unfortunately, not all treatments are suitable for long-term management, and selected patients may be ineligible for some options due to intolerability or contraindications. For example, patients with moderate-to-severe GIBD may require induction treatment with corticosteroids, but these should not be used longer-term due to risks and side effects [12]. Similarly, high corticosteroid doses may lead to gastrointestinal perforation. Thus, reduction or discontinuation should be a priority in patients with GIBD [36].

In IBD, a significant proportion of patients fail to respond to anti-TNF-α induction treatment (20–30% in patients with CD), have only a partial response, or become unresponsive over time (30–40% within one year of treatment) [67]. Loss of response to anti-TNF-α treatment has also been observed in patients with GIBD [18,37]. In patients who fail to respond to anti-TNF-α agents, the cause is often immunogenicity [67]. While immunogenic responses can be induced by all biological agents, it has been suggested that fusion proteins are less frequently associated with immunogenicity, impacting efficacy and safety [82]. Furthermore, GIBD is a disease marked by multiple relapses, and certain patients, e.g., those who undergo intestinal surgery, are at particularly high risk for recurrence. Treatments are needed to prevent the recurrence of the disease following intestinal surgery, or, if this is not possible, patients need alternative treatments to control the disease in later stages [6,7,12].

Although biological agents greatly improved treatment and disease control in patients with GIBD, not all respond to current anti-TNF-α agents. Those who do respond are more susceptible to bacterial and fungal infections due to the role of TNF-α in modulating the innate immune system [41,42].

In clinical trials involving patients with rheumatoid arthritis, tocilizumab, the only anti-IL-6R mAb currently under investigation for BD (including a few cases of GIBD), appeared to pose a risk of serious and opportunistic infections, similar to anti-TNF-α agents in clinical use [52,73]. Thus, preservation of the relevant processes mediated by classic IL-6 signaling, such as infection defense, intestinal regeneration, and tissue/wound healing, could be assumed upon an exclusive inhibition of IL-6 trans-signaling. Whether this would result in a lower risk of serious infections and/or gastrointestinal perforation in patients remains to be clarified [52]. Such safety aspects are of particular interest considering that patients with inflammatory diseases, including IBD and GIBD, often require long-term or even life-long treatment [51,52].

Blockade of IL-6 trans-signaling can be achieved at comparably low therapeutic concentrations and does not appear to hamper infection control or IL-6-mediated acute-phase reaction in preclinical models [83,84,85]. Maintaining classic IL-6 signaling might be important for cytoprotective responses in vascular endothelial cells, which have a key role in controlling tissue damage and repair [86]. Another preclinical study found that low-dose injections of IL-6 improved mesenteric perfusion and post-ischemic mucosal healing [87]. In a case report of a patient with GIBD, increased IL-6 expression after infliximab-induced remission was suspected to be partly responsible for the resolution of severe inflammation in the colon tissue [88]. This contribution of IL-6 to the remission of gastrointestinal lesions may be due to the promotion of epithelial regeneration, possibly facilitated by classic IL-6 signaling.

## 5. Expert Opinion

Although the incidence of BD is trending downwards, there is a high unmet need for adequate treatment of patients with severe disease progression and high relapse rate or patients who are refractory to conventional treatment or TNF-α inhibitors. Apart from elevated IL-6 levels found in patients with GIBD, currently available clinical evidence on the involvement of IL-6 in GIBD pathogenesis is scarce. Instead, the scientific rationale for the evaluation and implementation of IL-6 inhibitors in GIBD is mostly driven by the fact that IL-6 plays a key role in inflammatory and autoimmune processes and is supported by sparse yet promising clinical data in IBD. It should be noted that, despite the immunopathological similarities with IBD (particularly with CD) in terms of gastrointestinal involvement, the pathophysiology of GIBD is much broader and includes extraintestinal manifestations. An increased understanding of the pleiotropic physiological role of IL-6 would facilitate a better definition of the therapeutic spectrum of IL-6 inhibitors in GIBD. Yet, in our view, it is likely that scientific evidence will be derived directly from clinical observations in patients with IBD, (GI)BD, or both. The research on the therapy of IBD is already well advanced and has produced a considerable number of effective drugs, which form the basis for approved GIBD treatment options and those currently under investigation. The clinical effectiveness of IL-6 inhibitors in IBD could be an initial indication of their potential relevance in GIBD. Study results to date, particularly the effectiveness of olamkicept evaluated in two phase 2 trials in patients with IBD [71,72], are promising, but the benefits of IL-6 inhibitors need to be verified in larger studies to warrant their potential approval in this indication.

With regard to safety, overcoming the serious gastrointestinal events and infections observed with tocilizumab and PF-04236921 [68,70] remains the main concern in the future development of IL-6 inhibitors. According to preclinical studies, these appear to be attributable to the pleiotropic role of IL-6 in immunocompetence—more specifically, the inhibition of the classic signaling pathway [89]. Theoretically, selective inhibition of IL-6 trans-signaling could be expected to have a low impact on infection control and intestinal healing. In patients with IBD, olamkicept has already shown a more favorable safety profile than tocilizumab and PF-04236921 [71,72]. It remains to be seen whether and to what extent selective IL-6 trans-signaling inhibition is superior to its global counterpart in patients with GIBD, who generally have an increased risk of intestinal perforation.

Previous clinical experience with IL-6 inhibitors in patients with GIBD is limited to a few case reports on treatment with tocilizumab [75,76]. Initial clinical studies in patients with (GI)BD would be desirable. Key considerations for such future trials to evaluate the utility of IL-6 inhibitors are the selection of the patient population and suitable efficacy endpoints. The observed downward trend in the prevalence of BD in recent years, especially for severe cases, and the identification of patients with the gastrointestinal phenotype among the BD population may pose a challenge for sufficient recruitment. Similarly to other non-TNF-α biologics currently under investigation, IL-6 inhibitors may be well-positioned to be studied initially in patients who failed to respond to anti-TNF-α agents and/or corticosteroids—a subgroup with limited treatment options. Alternatively, as IL-6 inhibitors may serve as an alternative to anti-TNF-α treatment, they may be investigated in anti-TNF-α-naïve patients. Potential exclusion criteria of future clinical trials may be derived from the known clinical side effect profile of IL-6 inhibitors, e.g., patients with certain infections, active tuberculosis or malignancies, abnormal blood cell counts, and high risk of gastrointestinal perforation. In terms of clinical endpoints, it is proposed to assess both clinical and endoscopic gastrointestinal activity in patients, even though there are currently no standardized outcome measures for GIBD studies [90]. Like in IBD trials, induction of clinical remission and response, mucosal healing, and steroid-free remission may serve as efficacy endpoints in GIBD studies.

Our multi-disciplinary author group (i.e., rheumatologists and gastroenterologists) is confident that, in the evolving treatment landscape for GIBD, biologics that target cytokines involved in disease pathology, including IL-6 inhibitors, will play an increasingly important role in the coming years. Biological treatments are generally associated with high costs, but with approval in GIBD and insurance coverage (similar to TNF-α inhibitors), these treatments may achieve long-term remission in patients. They can thus positively influence total costs over the course of the disease by reducing the likelihood of multiple surgeries and side effects, e.g., from repeated steroid use. Once we have a better understanding of the genetic and/or molecular pathological factors of the disease and potential biomarkers for GIBD, companion diagnostics for targeted treatments would also be valuable in the future to tailor treatment to the patient.

In our view, the selective inhibition of IL-6 trans-signaling will bring new impetus to the development of this drug class, particularly regarding safety outcomes. With the aforementioned challenges in mind, we hope that appropriate pilot studies can soon be initiated to provide the necessary clinical evidence for the future development of IL-6 inhibitors for patients with GIBD.

## 6. Conclusions

A high unmet need for the management of BD persists, particularly for phenotypes with higher mortality rates, such as GIBD. Effective IBD treatments typically form the basis for the development of drugs to treat GIBD. IL-6 plays a key role in the trans-signaling pathway that triggers pro-inflammatory processes. The exclusive inhibition of IL-6 trans-signaling offers a hitherto unique mode of action that may preserve the anti-inflammatory activity of IL-6, such as its contribution to intestinal tissue regeneration and infection defense. Although further evidence is needed, available clinical data seem promising, suggesting that such an exclusive mode of action may lead to alternative treatment options in patients with IBD and potentially GIBD.

## Figures and Tables

**Figure 1 biomedicines-13-00247-f001:**
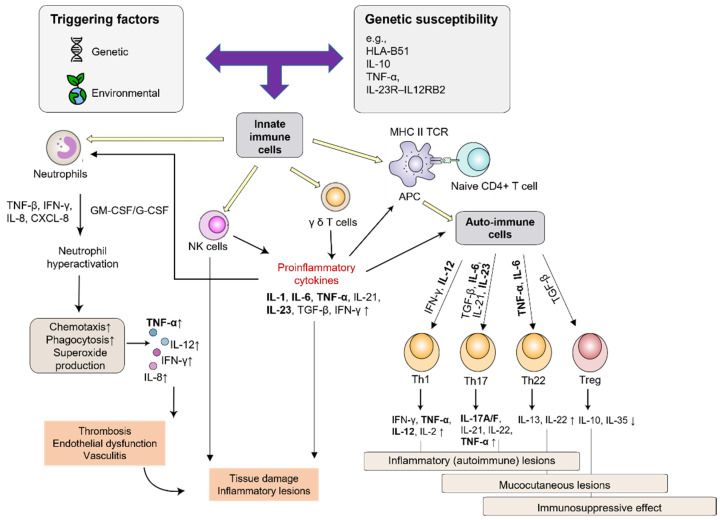
Innate and adaptive immune responses in Behçet’s disease [20]. Cytokines targeted by approved and emerging biological agents appear in bold. Adapted from Tong et al. [20], published under the Creative Commons Attribution License (CC BY). APC, antigen-presenting cell; CXCL-8, interleukin-8; G-CSF, granulocyte-colony stimulating factor; GM-CSF, granulocyte macrophage colony-stimulating factor; HLA, human leukocyte antigen; IFN, interferon; IL, interleukin; MHC, major histocompatibility complex; NK, natural killer; TCR, T-cell receptor; TGF, transforming growth factor; Th, T helper cell; TNF, tumor necrosis factor; Treg, regulatory T-cell.

**Figure 2 biomedicines-13-00247-f002:**
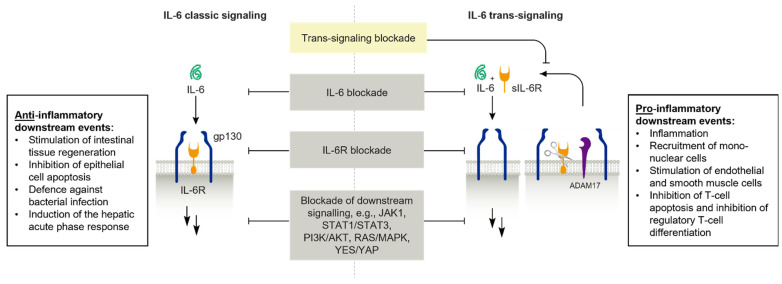
Targets of the classic and trans-signaling pathways of IL-6 [51,52]. Adapted and reprinted with permission from Rose-John et al. [52]. ADAM, A disintegrin and metalloprotease; gp, glycoprotein; IL, interleukin; IL-6R, interleukin-6 receptor; JAK, Janus kinase; MAPK, mitogen-activated protein kinase; PI3K, phosphoinositide 3-kinase; sIL-6R, soluble IL-6R; STAT, signal transducer and activator of transcription; RAS, Rat sarcoma; YAP, yes-associated protein.

**Table 1 biomedicines-13-00247-t001:** Gastrointestinal Behçet’s disease (GIBD) diagnosis according to the Korean Inflammatory Bowel Disease group [10].

Endoscopic	Clinical ^2^	GIBD
Typical ^1^	Systemic BD	Definitive
	Only oral ulcer	Probable
	No finding	Suspected
Atypical	Systemic BD	Probable
	Only oral ulcer	Suspected
	No finding	Non-diagnostic

^1^ ≤5 Ulcers, oval or round, deep, discrete borders. ^2^ According to the diagnostic criteria of the Research Committee of Japan, complete, incomplete, and suspected subtypes of systemic BD were classified. BD, Behçet’s Disease; GIBD, gastrointestinal BD.

**Table 2 biomedicines-13-00247-t002:** Clinical study outcomes for anti-TNF-α agents (i.e., infliximab and adalimumab) in gastrointestinal Behçet’s disease (GIBD).

	Indication	Number of Patients, Total and per Group	Efficacy Endpoint	Efficacy Outcome	Safety Outcome
Infliximab (IFX), induction and maintenance treatment Approvals: ^†^ for CD and UC in Australia, Canada, Israel, South Africa, the US, and select European and Asian countries; for BD in Japan					
Prospective, single-arm, open-label study (phase 3) (Hibi et al., 2016) [37]	Mild-to-severe BD (including GIBD, NBD, VBD)	N = 18 (3 types of BD) NBD, n = 3 VBD, n = 4 GIBD, n = 11	Clinical response at week 30 (no clinical symptoms with healed ulcer)	IFX 5 mg/kg: 55%	Infections (n) All BD (11), GIBD (7) Including upper respiratory tract infection (5), nasopharyngitis (4), gastroenteritis (2), infectious enteritis (2) No serious infections No AEs leading to drug discontinuation Serious AEs (n) All BD (2), GIBD (2) Worsening of the underlying disease and cataracts (1 GIBD patient); no causal relationship with IFX treatment
BEGIN single-arm, open-label study (phase 3) (Cheon et al., 2023) [38]	Moderate-to-severe GIBD, refractory to conventional treatment	N = 33	Clinical response (≥20-point decrease in DAIBD) at weeks 14 and 32	IFX 5 mg/kg: 92%	Infections (n), all not related to IFX Induction phase (3), maintenance phase (6) Including nasopharyngitis, anal abscess, and cystitis No serious AEs related to IFX No IFX-related AEs leading to discontinuation
Adalimumab (ADA), induction and maintenance treatment Approvals: ^†^ For CD and UC in Australia, Brazil, Canada, Mexico, New Zealand, South Africa, Switzerland, the US, and select European and Asian countries; for BD in Israel, Japan, and South Korea					
Prospective, single-arm, open-label study (phase 3) (Tanida et al., 2015) [39]	GIBD, refractory to corticosteroid and/or immunomodulator therapies	N = 20	Clinical response (according to present GI symptoms and ulcer size) at weeks 24 and 52	Clinical response ADA 40 mg: 45% and 60%	No new safety signals were observed, no deaths, no cases of malignancy, congestive heart failure, demyelination, or lupus-like syndrome AEs possibly related to ADA (n) Week 20 (5), week 52 (5) Infections (n) Week 20 (9), week 52 (14) Some patients (4) experienced an infection after dose escalation
Prospective, observational study (Suzuki et al., 2021) [40]	GIBD	N = 462	Clinical response (≥50% reduction in size of largest ulcer) at weeks 0–24 and 104–156	ADA 160 mg and 80 mg (at weeks 0–24): 47% ADA 40 mg (at weeks 104–156): 68%	ADRs and serious ADRs in 120 and 51 patients; incidence is significantly higher in patients with comorbidities and concomitant corticosteroids Infections (n) Any (47), serious (18) Pulmonary TB related to ADA (1); TB probably related to ADA (2)
			Symptom-free or symptoms did not affect daily life at weeks 52 and 104	ADA 40 mg (at weeks 52 and 104): 85% and 88%	

^†^ According to http://www.globaldata.com (accessed on 10 January 2025). ADA, adalimumab; ADR, adverse drug reaction; AE, adverse event; BD, Behçet’s disease; CD, Crohn’s disease; DAIBD, disease activity index for intestinal BD; GI, gastrointestinal; GIBD, gastrointestinal BD; IFX, infliximab; NBD, neurological BD; TB, tuberculosis; TNF, tumor necrosis factor; UC, ulcerative colitis; US, United States; VBD, vascular BD.

**Table 3 biomedicines-13-00247-t003:** Clinical study outcomes for agents targeting IL-6 signaling (i.e., tocilizumab, PF-04236921, and olamkicept) in inflammatory bowel disease.

Study (Author, Year)	Inflammatory Bowel Disease	Number of Patients, Total and per Group	Efficacy Endpoint	Efficacy Outcome	Safety Outcome
Tocilizumab (TCZ), 12-week treatment In development; no approvals for IBD ^†^					
Pilot placebo-controlled study (Ito et al., 2004) [66]	Active CD	N = 36 13 (TCZ 8 mg/kg, Q4W) 10 (TCZ 8 mg/kg, Q2W) 13 (placebo)	Clinical response at week 12 (CDAI-70)	TCZ 8 mg/kg, Q4W: 42% TCZ 8 mg/kg, Q2W: 80% Placebo: 31%	Serious AEs (n) in TCZ groups Paralytic ileus (1, discontinued TCZ) and abdominal pain/gastrointestinal bleeding (3, possibly related to TCZ) No serious infections
PF-04236921 (PF), induction (days 1 and 28), and maintenance treatment In development; no approvals for IBD ^†^					
ANDANTE I (placebo-controlled) ANDANTE II (open-label extension) (Danese et al., 2019) [68]	Moderate-to-severe CD, refractory to anti-TNF-α treatment	N = 249 68 (PF 10 mg) 71 (PF 50 mg) 70 (placebo) 40 (PF 200 mg), discontinued	Response at week 8 or 12 (CDAI-70)	PF 10 mg: no significant improvement compared with placebo PF 50 mg: 49% and 47% Placebo: 31% and 29%	For both studies, most frequent TEAEs and serious AEs were CD-related (including worsening, exacerbation, and flare of CD and abdominal pain) and nasopharyngitis Serious AEs (n) in PF groups ANDANTE I (27), ANDANTE II (58) Serious GI (n) events in PF groups ANDANTE I (6), ANDANTE II (10)
			Response and remission at week 48 (exploratory endpoint, ANDANTE II)	PF 50 mg: 40% and 32%	
Olamkicept (OLA), 12-week treatment In development; no approvals for IBD ^†^					
FUTURE Exploratory study (phase 2a) (Schreiber, et al. 2021) [71]	Moderate-to-severe CD or UC	N = 16 9 (UC, OLA 600 mg, Q2W) 7 (CD, OLA 600 mg, Q2W)	Clinical remission at week 14 (primary assessment; Mayo ≤ 2, bleeding of 0, endoscopy ≤ 1 for UC; CDAI < 150 for CD)	OLA, UC: 22% OLA, CD: 14%	Note: The study was not powered to firmly assess safety AEs in 13 patients, but unspecific, unrelated to drug exposure, not indicative of severe immune suppression Serious AEs (n) unlikely to be related to OLA Serious AEs (5), e.g., atrial fibrillation, unspecific weakness No serious infections
			Clinical and endoscopic response at week 14 (reduction in Mayo ≥ 3 points, bleeding score ≤ 1 for UC; reduction in CDAI >100 for CD)	OLA, UC: 55% and 55% OLA, CD: 28% and 14% Note: The study was not designed to assess the efficacy	
Randomized, double-blind, placebo-controlled study (phase 2) (Zhang et al., 2023) [72]	Active moderate-to-severe UC	N = 91 31 (OLA 300 mg, Q2W) 30 (OLA 600 mg, Q2W) 30 (placebo)	Clinical and endoscopic response at week 14 (reduction in Mayo ≥ 3 points, bleeding score ≤ 1 for UC; reduction in CDAI >100 for CD)	OLA 300 mg: 43% OLA 600 mg: 59% Placebo: 35%	Drug-related TEAEs (n) 600 mg OLA (16), 300 mg OLA (18), placebo (15) Most common with OLA were bilirubin in urine (7), hyperuricemia (5), and increased AST levels (4) Drug-related infections (n) in OLA groups 600 mg (2), 300 mg (3) Positive IFN-γ release assay (5), no tuberculosis infection diagnosed
			Clinical remission (Mayo ≤ 2, bleeding of 0, endoscopy ≤ 1 for UC; CDAI <150 for CD) and mucosal healing (endoscopic subscore 0/1)	OLA 300 mg: 7% and 10% OLA 600 mg: 21% and 35% Placebo: 0% and 3%	

^†^ According to www.globaldata.com (accessed on 10 January 2025). AE, adverse event; anti-TNF-α, anti-tumor necrosis factor-α; AST, aspartate aminotransferase; CD, Crohn’s disease; CDAI, Crohn’s Disease Activity Index; IBD, inflammatory bowel disease; IFN, interferon; OLA, olamkicept; PF, PF-04236921; Q4W, every 4 weeks. Q2W, every two weeks; TCZ, tocilizumab; TEAE, treatment-emergent adverse event; UC, ulcerative colitis.

## Data Availability

Data sharing is not applicable to this article as no new data were created or analyzed.

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
