# Peer review of "Assessment of IL-6 Pathway Inhibition in Gastrointestinal Behçet’s Disease from Immunological and Clinical Perspectives"

_biomedicines, 2025, doi:10.3390/biomedicines13010247_

Round 1

Reviewer 1 Report

Comments and Suggestions for Authors

The manuscript “Assessment of IL-6 pathway inhibition in gastrointestinal Behçet’s disease from immunological and clinical perspectives” is good piece of work that have explained the potential of blocking IL-6 as a treatment for GIBD which is a serious intestinal problem. There are some suggestions for improvements

1.      While geography is mentioned as influencing the prevalence of GIBD, the discussion doesn’t explore the reasons behind this, such as diet, environmental exposures, or healthcare access.

2.      The section “Overview of Approved and Emerging Treatments for GIBDcovers a broad range of treatments (e.g., 5-ASA, anti-TNF-α agents and corticosteroids). However, it lacks consistent depth. The section provides relatively less information about the emerging therapies like baricitinib and apremilast.

3.      The emerging treatments, such as anti-IL-1 agents and anti-IL-6 monoclonal antibodies are not discussed in much detail. Ongoing clinical trials??

4.      In the treatment efficacy, the cost, accessibility, and affordability of treatments especially for the newer agents need to be discussed.

5.   What would be the role of personalized or precision medicine in GIBD treatment. How can the genetic factors or patient-specific characteristics guide treatment choices?

Author Response

Comments 1: While geography is mentioned as influencing the prevalence of GIBD, the discussion doesn’t explore the reasons behind this, such as diet, environmental exposures, or healthcare access.

Response 1: We agree with the reviewer that a little more context, i.e. the relevance of genetic and environmental factors, would be helpful. However, the aetiopathogenesis of BD is still largely unknown and the relevance of specific diets or exposures remain unclear.

Regarding GIBD, we would like to emphasize that the prevalences are more “observations” from respective studies that do not necessarily reflect the true geographic variation due to (partly methodological) reasons given on page 2 lines 80–84.

Accordingly, we have included the following statement in the introduction section (page 2, lines 66–69):

In addition to genetic susceptibility (human leukocyte antigen B51 [HLA-B51], cytokines etc.), environmental factors (e.g., infectious factors, dysbiosis) that trigger immune responses in BD are suggested to contribute to these geographic prevalence trends [4,5]

Furthermore, we have made the following updates in subsection 2.1 (page 2, lines 79–80): “Patients with GIBD are more frequently diagnosed in distinct geographic regions and the reported global prevalence varies significantly [4].

Comments 2: The section “Overview of Approved and Emerging Treatments for GIBD” covers a broad range of treatments (e.g., 5-ASA, anti-TNF-α agents and corticosteroids). However, it lacks consistent depth. The section provides relatively less information about the emerging therapies like baricitinib and apremilast.

Response 2: Implemented. We have added the following statements on baricitinib and apremilast (page 7, lines 285–292). We have however kept this brief due to the limited relevance of these two drugs in GIBD to date.

“Other suggested treatment options for GIBD include the small molecule drugs baricitinib, a Janus kinase (JAK)1/JAK2 inhibitor, and the phosphodiesterase-4 inhibitor apremilast [47]. Baricitinib has shown favourable and glucocorticoid-sparing effects in a few patients with refractory GIBD [46]. Apremilast is approved for the treatment of oral ulcers associated with BD, but has also been shown to improve gastrointestinal manifestations in some cases, either alone or in combination with anti-TNF-α agents [48,49]. Diarrhea is one of the most common AEs of apremilast,[50] which must be taken into account when considering its use in patients with GIBD. Overall, […]”

Comments 3: The emerging treatments, such as anti-IL-1 agents and anti-IL-6 monoclonal antibodies are not discussed in much detail. Ongoing clinical trials?

Response 3: We are not aware of any ongoing trials of anti-IL-1 agents in patients with GIBD. As described on page 7, there is little information on the potential use of anakinra and canakinumab in patients with GIBD.

We have made the following updates to clarify the current situation of the anti-IL-1 agents anakinra and canakinumab for patients with GIBD (page 7, lines 276–279):

They may have potential in the management of GIBD, but these assumptions are based on observations in small subpopulations of patients with gastrointestinal symptoms from two case series reports and one retrospective cohort study [45].”

In addition, we have slightly edited the points on tocilizumab in this section (page 7, lines 279–281), as tocilizumab is covered in more detail in section 4.2.

Controversial observations have been made regarding the efficacy of the anti-IL-6 monoclonal antibody (mAb) tocilizumab in GIBD (discussed in greater detail in the following sections), and further investigation is required due to a risk of intestinal perforation [45,46]”

Comments 4: In the treatment efficacy, the cost, accessibility, and affordability of treatments especially for the newer agents need to be discussed.

Response 4: Implemented. We updated several sections to include these topics and their relevance to patients with GIBD.

We have made the following addition on page 7, line 250:

“For example, infliximab and adalimumab (both of which are approved for GIBD and covered by insurance in Japan) have […]”

In addition, we included a statement in section 5 (Expert Opinion; page 15, line 542–546)

“Biological treatments are generally associated with high costs, but with approval in GIBD and insurance coverage (similar to TNF-α inhibitors), these treatments may achieve long-term remission in patients. They can thus positively influence total costs over the course of the disease by reducing the likelihood of multiple surgeries and side effects, e.g., from repeated steroid use.”

Comments 5: What would be the role of personalized or precision medicine in GIBD treatment. How can the genetic factors or patient-specific characteristics guide treatment choices?

Response 5: We agree with the reviewer that the idea of precision medicine is of interest in this field in the future. However, there is at present fairly limited knowledge on the pathological mechanisms of GIBD.

We have however added the following text to provide some perspective around this interesting topic (pages 15–16, lines 546–549):

“Once we have a better understanding of the genetic and/or molecular pathological factors of the disease, and potential biomarkers for GIBD, companion diagnostics for targeted treatments such as IL-6 inhibitors would also be valuable in the future to tailor treatment to the patient.” 

Reviewer 2 Report

Comments and Suggestions for Authors

The review is well-written and provides a comprehensive review of the role of IL-6 inhibitors in GIBD. I have some minor comments:

1)      In the abstract please mention the inhibitors of IL6

2)      The English in Simple Summary is poor or not very scientific sometimes. For example:

…can also cause a hole in the gut, which can be life-threatening.

3)      In the keywords, consider replacing "signal pathways" with "signaling pathways"

4)      In line 97 please correct this: Error! Reference 97 source not found.T

Author Response

Comments 1: In the abstract please mention the inhibitors of IL6.

Response 1: Implemented as follows:

“Given the current paucity of data, we reflected on the potential of IL‑6 inhibitors such as tocilizumab and olamkicept […]”

Comments 2: The English in Simple Summary is poor or not very scientific sometimes. For example: …can also cause a hole in the gut, which can be life-threatening.

Response 2: We would like to emphasize that this section is intended for general readers and detailed scientific terminology was therefore deliberately avoided to improve accessibility. The section was checked by native English-speakers during revision. We have also amended the cited statement to “…cause a hole (perforation) in the gut” to clarify it; we hope this addresses the reviewer’s point.

Comments 3: In the keywords, consider replacing "signal pathways" with "signaling pathways"

Response 3: We have made the update to “signaling pathways” as suggested.

Comments 4: In line 97 please correct this: Error! Reference 97 source not found.

Response 4: Many thanks, implemented. 

Reviewer 3 Report

Comments and Suggestions for Authors

This paper presents an interesting update on the role of IL6 pathway inhibition and Gastrointestinal Behçet’s Disease (GBD) course. 

The paper very well reviews the pathophysilogy of GBD as well as gives nice overview of IL6 pathway in general. It also gives a good amount of information about the biochemistry of IL6. The paper summarizes the interconnections between GBD and IL6 in both, bench and clinical science. I feel that the paper has very balanced and critical perspective what is truly important for the review papers.  On the other hand, I would like to suggest some improvements in this paper to increase its quality and an interest to the readers. Please follow my majors: 

1) It would be helpful to include the comparison of the effects of IL6 inhibtion with other key components of the anti-inflammatory pathways in GBD, like anti-TNFalpha or so. 

2) Another worth mentioning thing is to include information on how the patient's heterogeneity (genetic, geographic, life style or phenotypic differences in GIBD) might impact the uasge of these inhibtiors in GBD.

3) The Auhtors should put weight on the adverse effects of these therapeutics, since it is well known that not everyone tolerates well these drugs. 

4) Please highlight how this treatment might be integrated into existing therapeutic patterns in treatment of GBD.

5) Please also comparethe safety profiles between broad-spectrum IL-6 inhibitors and selective trans-signaling blockers like olamkicept.

6) Please try to refresh the information in the manuscript by adding new clinical trials that are in the stage of recruiting/active phase.

once these changes are made, I would be happy to see the revised version. 

Best. 

Author Response

Comments 1: It would be helpful to include the comparison of the effects of IL6 inhibition with other key components of the anti-inflammatory pathways in GIBD, like anti-TNFalpha or so.

Response 1: Implemented as described below. We hope the reviewer feels our approach of primarily using Figure 1 to illustrate the innate and adaptive immune responses in BD and the suggested role of different biologics in anti-inflammatory pathways in (GI)BD sufficiently addresses their point. Supporting aspects on how anti-IL-6 and anti-TNF-α agents might come into play in (GI)BD are given on page 4, second paragraph in section 2.3 and page 6, second paragraph in section 2.4.

The following additions have been made to further clarify to readers where the relevant information can be found (page 4, lines 163—164 and lines 167–170, respectively):

“Mechanisms and pivotal cytokines that stimulate T-cell differentiation towards pro-inflammatory subsets are shown in Figure 1”

“Furthermore, IL-6 and TNF-α enable differentiation into Th22 cells, which produce IL-22 – this is another type of pro-inflammatory cytokine that particularly contributes to mucocutaneous lesions in patients with BD”

Comments 2: Another worth mentioning thing is to include information on how the patient's heterogeneity (genetic, geographic, lifestyle or phenotypic differences in GIBD) might impact the usage of these inhibitors in GIBD.

Response 2: Partially implemented. It is not yet possible to estimate the effect of patient heterogeneity on the response to these inhibitors, but we agree that tailoring targeted therapies such as IL-6 inhibitors to individual patients is a task for the future. We have added the following text to provide some perspective around this topic (pages 15, lines 546–549): “Once we have a better understanding of the genetic and/or molecular pathological factors of the disease, and potential biomarkers for GIBD, companion diagnostics for targeted treatments such as IL-6 inhibitors would also be valuable in the future to tailor treatment to the patient.”

Comments 3: The Authors should put weight on the adverse effects of these therapeutics, since it is well known that not everyone tolerates well these drugs. 

Response 3: This is a good point and one that we have implemented. We have added some text on anti-TNF-α agents to supplement the issues and concerns regarding this topic provided in sections 3, 4.1 and 4.2 (including tables) as well as page 15 (line 522-531).

Adverse events (AEs) include autoimmune and cardiac conditions, malignancies, and infections (mild or serious), the latter being a significant concern and especially in the context of long-term use [44].” (page 7, lines 258—260)

Comments 4: Please highlight how this treatment might be integrated into existing therapeutic patterns in treatment of GIBD.

Response 4: We thank the reviewer for this comment and partially implemented it. The revised paragraph on potential target groups for future clinical studies in section 5 (page 15, lines 522-531) contains an estimate of where IL-6 inhibitors might fit into existing GIBD treatment patterns.

We have added the following additional point (page 15, line 529–531):

“Alternatively, as IL-6 inhibitors may serve as an alternative to anti-TNF-α treatment, they may be investigated in anti-TNF-α-naïve patients.”

Comments 5: Please also compare the safety profiles between broad-spectrum IL-6 inhibitors and selective trans-signaling blockers like olamkicept.

Response 5: We agree that safety is a key element in comparing selective inhibition of IL‑6 trans‑signaling and global IL-6 inhibition. Together with reporting current findings on the clinical safety of IL-6, we have referred to potential advantages/disadvantages of these two mechanisms of action throughout the manuscript (e.g., on page 13, last paragraph in section 4.1, and page 15, second paragraph in section 5)  

We hope the reviewer feels the explanations are sufficient in view of the currently limited clinical data on IL-6 inhibitors in patients with GIBD.

Comments 6: Please try to refresh the information in the manuscript by adding new clinical trials that are in the stage of recruiting/active phase.

Response 6: Partially implemented. We did not find any new drug intervention trials on GIBD in the recruitment/active stage, but nevertheless took the opportunity to add the following insights to different sections as a refresher:

“Additional insights are expected from an ongoing clinical trial that aims to further elucidate the pathogenesis of GIBD [31].” (page 6, lines 213–214)

“More recently, the real-world efficacy and safety of infliximab in patients with GIBD and other subtypes have been confirmed [43].” (page 7, lines 256–258)

“Data from two phase 1 trials demonstrate safety and tolerability after single and multiple dosing of olamkicept in healthy subjects and patients with CD [71].” (page 14, lines 414–416)

Reviewer 4 Report

Comments and Suggestions for Authors

In general, the author has fully understood and expressed the research in this field, but the writing of the article uses few pictures, which is not intuitive enough. It is hoped that the author can add some pictures to express the methods in the article and use pictures to express the comparison of data.

Author Response

General Comment: In general, the author has fully understood and expressed the research in this field, but the writing of the article uses few pictures, which is not intuitive enough. It is hoped that the author can add some pictures to express the methods in the article and use pictures to express the comparison of data.

Response: We thank the reviewer for their assessment and are very glad that they formed a positive impression, but would prefer not to implement this suggestion. Since our review article takes a narrative and not a systematic literature search approach, we are limited in the visual presentation of our methods.

Regarding the data shown for a variety of drugs, we feel that the tables we provided are the most efficient way to present the data in a structured and comprehensive way, and we already have two illustrative figures, which is not unusual for a review of this length.

Round 2

Reviewer 1 Report

Comments and Suggestions for Authors

The authors have improved the manuscript and can be accepted for publication

Reviewer 3 Report

Comments and Suggestions for Authors

The Authors present a revised version of their paper. After carefully re-evaluating the draft, I have to admit that most of my majors were addressed correctly as well as broadly discussed. Since the paper is a review article, I do not have any further comments or points to be raised. I am satisfied with the peer-review process, and I am supportive of acceptance of the paper.